# Production of a Fungal Punicalagin-Degrading Enzyme by Solid-State Fermentation: Studies of Purification and Characterization

**DOI:** 10.3390/foods12040903

**Published:** 2023-02-20

**Authors:** Pedro Aguilar-Zárate, Gerardo Gutiérrez-Sánchez, Mariela R. Michel, Carl W. Bergmann, José J. Buenrostro-Figueroa, Juan A. Ascacio-Valdés, Juan C. Contreras-Esquivel, Cristóbal N. Aguilar

**Affiliations:** 1Engineering Department, Instituto Tecnológico de Ciudad Valles, Tecnológico Nacional de México, Ciudad Valles, San Luis Potosí C.P. 79010, Mexico; 2Complex Carbohydrate Research Center, The University of Georgia, Athens, GA 30602, USA; 3Research Center in Food and Development A. C. Cd., Delicias, Chihuahua C.P. 33089, Mexico; 4Bioprocesses Research Group, Food Research Department, School of Chemistry, Universidad Autónoma de Coahuila, Saltillo, Coahuila C.P. 25280, Mexico

**Keywords:** ellagitannins, ellagic acid, *Aspergillus niger*, solid-state fermentation, enzyme kinetics

## Abstract

The present work describes the purification of an enzyme capable of degrading punicalagin. The enzyme was produced by *Aspergillus niger* GH1 by solid-state fermentation, and the enzyme production was induced by using ellagitannins as the sole carbon source. The purification steps included the concentration by lyophilization, desalting, anionic exchange, and gel filtration chromatography. The enzyme kinetic constants were calculated by using punicalagin, methyl gallate, and sugar beet arabinans. The molecular mass of the protein was estimated by SDS-PAGE. The identified bands were excised and digested using trypsin, and the peptides were submitted to HPLC-MS/MS analysis. The docking analysis was conducted, and a 3D model was created. The purification fold increases 75 times compared with the cell-free extract. The obtained *Km* values were 0.053 mM, 0.53% and 6.66 mM for punicalagin, sugar beet arabinans and methyl gallate, respectively. The optimal pH and temperature for the reaction were 5 and 40 °C, respectively. The SDS-PAGE and native PAGE analysis revealed the presence of two bands identified as α-l-arabinofuranosidase. Both enzymes were capable of degrading punicalagin and releasing ellagic acid.

## 1. Introduction

Ellagitannins are secondary metabolites obtained from plants and are one of the major classes of polyphenols [1]. The monomeric unit of ellagitannins possesses a hexahydroxydiphenic (HHDP) acid esterified generally to a glucose core. Ellagitannins can be defined in a narrow sense as HHDP esters of carbohydrates, while in a wider sense, they include compounds derived from oxidative transformations [2].

Punicalagin is an ellagitannin obtained mainly from pomegranate (*Punica granatum*) and is a potent antioxidant. The biological activity of punicalagin depends on its ability to hydrolyze into ellagic acid by the cleavage of ester bonds [3]. When the HHDP group is cleaved from the sugar core, the parent acid rapidly lactonizes to yield the dilactone ellagic acid [4]. Ellagic acid possesses a wide range of biological activities promoting good human health, thus calling the attention of consumers. Antioxidant function, estrogenic and/or antiestrogenic, anti-inflammatory and prebiotic effects are examples of biological applications of ellagic acid [5].

The hydrolysis of ellagitannins into monomeric products has been carried out by acids and microbial enzymes [6]. The use of biotechnological tools (such as solid-state fermentation) to produce microbial enzymes capable of releasing ellagic acid is a feasible alternative [7].

The microbial enzymes involved in ellagitannins hydrolysis have been a discussion topic. Particularly, it has been reported that microbial tannase is active on galloyl residues as well as hexahydroxydiphenoyl and other residues of ellagitannins [6]. In addition, it has been mentioned that this enzyme is not active against all hydrolysable tannins. Many other enzymes, including tannase, phenol oxidase, and decarboxylase, are the enzymes involved in the degradation of gallotannins and ellagitannins [8].

Some other authors have reported an enzyme called ellagitannin acyl hydrolase (EAH) or ellagitannase as the enzyme able to hydrolyze ellagitannins [9,10,11,12,13,14,15]. In addition, the valonia tannin hydrolase obtained from *Aspergillus niger* SHL6 has been reported as the enzyme involved in the degradation of valonia tannins [16]. Many of the authors have reported the use of *Aspergillus niger* GH1 as a model to produce the ellagitannin-acyl hydrolyze enzyme [9,10,12,13]. All the previous reports used the crude enzymatic extract obtained from the fermentation of ellagitannins and demonstrated that it possesses some other activities different from tannase and ellagitannase: for example, polyphenol oxidase, and some glycosyl hydrolases such as β-glucosidase, cellulase, and xylanase [17,18]. Some glycosyl hydrolase enzymes possess a wide similarity and a variety of enzymatic functions and are classified into different families [19].

Therefore, the present study was aimed due to the lack of clear information related to the enzyme responsible for degrading ellagitannins. Our group has been working in the search for the EAH produced by *A. niger* GH1 and has proposed an enzyme different from tannase with the ability to degrade specifically ellagitannins. For that reason, the present work was conducted to purify, identify, and characterize the fungal enzyme capable of degrading ellagitannins using punicalagin as a substrate and *A. niger* GH1 as a model microorganism.

## 2. Materials and Methods

### 2.1. Cell Culture of Fungal Strain

The *A. niger* GH1 strain was obtained from the Food Research Department, University of Coahuila. The microorganism was previously tested as a tannin degrader [20]. The fungal strain was cryopreserved at −20 °C in media composed of glycerol and skim milk. The spores of *A. niger* GH1 were reactivated in potato-dextrose agar medium (Bioxon) at 30 °C for five days. The spores were harvested with a sterile solution of 0.01% of Tween 80 and counted in a Neubauer chamber.

### 2.2. Solid-State Fermentation (SSF) Conditions

SSF was carried out using high-density polyurethane foam (PUF) (Expomex, Mexico) as an inert support. PUF was ground to the particle size of 0.42–0.85 mm diameter and washed several times as reported by Mussatto et al. [21]. Czapek-Dox mineral medium modified according to Aguilar-Zárate et al. [9] was used and supplemented with 7.5 g/L of pomegranate–husk ellagitannins as a carbon source and inducer of EAH activity (provided by Food Research Department, Universidad Autónoma de Coahuila). The salt composition was 6.04 g/L KCl, 3.04 g/L MgSO_4_, 3.00 g/L NaNO_3_, and 1.00 g/L K_2_HPO_4_. PUF was impregnated with culture media including spore solution to reach 70% of moisture content. Fermentation was carried out at 30 °C and stopped after 18 h of culture. Fermented extracts were obtained by compressing the foam with sterile syringes; then, the liquid was filtered (Whatman 41) and the clear filtrate was assayed for EAH activity and protein content.

### 2.3. Ellagitannin-Acyl Hydrolase Assay

Ellagitannase activity was assayed according to Aguilar-Zárate et al. [9] and Buenrostro-Figueroa et al. [22]. Punicalagin anomers (purity ≥ 98%) were obtained as was reported by Aguilar-Zárate et al., [23] and they were used as enzyme substrate (1 mg mL^−1^ in 50 mM citrate buffer pH 5). Punicalagin anomers release ellagic acid when they are subjected to hydrolysis. An enzyme treatment control (1000 µL punicalagin + 50 µL 50 mM citrate buffer pH 5), an extract treatment control (1000 µL of 50 mM citrate buffer pH 5 + 50 µL of enzymatic extract), and the reaction mixture (1000 µL punicalagin + 50 µL to enzymatic extract) were prepared. All enzymatic preparations were allowed to react for 10 min at 60 °C in a water bath (Sheldon manufacturing m. 1225). The reaction was terminated by adding 1500 µL of absolute ethanol. Then, samples were sonicated in an ultrasonic water bath (Branston 2800) for 25 min, filtered through 0.45 µm membrane units (Millex^®^), and collected in vials. Ellagic acid quantification was carried out by HPLC (High-Performance Liquid Chromatography) equipment (Agilent system) at 254 nm. Then, 20 µL of samples was manually injected onto a C18 Denali column (250 mm × 4.6 mm, 5 µm, Grace, USA); the eluent was a gradient of aqueous formic acid (0.2%, *v*/*v*; solvent A) and acetonitrile (solvent B). The flow rate was maintained at 1.2 mL/min, and the elution of ellagic acid was monitored at 254 nm. The following gradients were used: initial, 3% B; 0–15 min, lineal 0–30% B. Then, the column was washed and reconditioned to initial conditions. Ellagic acid (0–100 ppm) (Sigma-Aldrich, St. Louis, MI, USA) standard solution was prepared for the calibration curve. One ellagitannase enzymatic unit was defined as the enzyme amount needed to release 1 µmol of ellagic acid per minute under the assay conditions.

### 2.4. Protein Assay

The protein concentration in the crude extract was quantified using a microplate reader (680, Bio-Rad, Hercules, CA, USA) by the Bradford Bio-Rad assay (Bio-Rad, Hercules, CA, USA) and the bicinchoninic acid (BCA) protein assay kit (23280, Thermo Scientific, Waltham, MA, USA). Bovine serum albumin (Sigma-Aldrich) was used as the protein standard in both methods.

### 2.5. Enzyme Purification

The crude enzyme extract (100 mL) was first filtrated using Whatman filter paper (No.1041) and 0.45 µm nylon membrane and then was frozen at −20 °C and freeze-dried. Desalting by gel filtration chromatography: Lyophilized crude extract was then injected onto a Hi Prep G25 Sephadex column (Marshal Biosciences) to separate protein, peptides, salts, and polyphenols. The elution was carried out using 50 mM pH 5 citrate buffer recovering 5 mL per fraction. Fractions with EAH activity were pooled, frozen at −80 °C, and lyophilized.

Anion exchange chromatography: Five hundred microliters of desalinized fractions were placed on a Macro-Prep High Q cartridge (Bio-Rad; Hercules, CA, USA). The column was equilibrated with 50 mM sodium acetate buffer pH 5 (buffer A). The enzyme was eluted using a linear gradient from 0% to 100% with 50 mM sodium acetate buffer pH 5 containing 1 M NaCl (buffer B) for 60 min. Thirty-three fractions were analyzed for EAH activity. Fraction identified with EAH activity was frozen at −80 °C, lyophilized and molecular characterized.

Gel filtration chromatography: The sample containing 200 µg of protein purified in the previous step was loaded onto a Superdex 75 (10 mm × 300 mm) gel filtration column. The column was pre-equilibrated with 50 mM sodium acetate pH 5 containing 150 mM NaCl and the protein was eluted using the same buffer at 0.5 mL/min. Fractions obtained were analyzed for EAH activity and molecular characterization.

### 2.6. Characterization of Enzyme

#### 2.6.1. Molecular Weight

The molecular weight of EAH was determined by sodium dodecyl sulfate-polyacrylamide gel (SDS-PAGE) according to Laemmli [24]. One hundred micrograms of lyophilized samples obtained in every purification step was dissolved in 25 µL of Laemmli sample buffer and heated at 95 °C for 5 min. Samples were loaded into a 12% SDS-PAGE precast gel (NuPAGE Novex 12% Bis-Tris, Life Technologies, Carlsbad, CA, USA) and separated using a Novex Mini-Cell (Invitrogen, Waltham, MA, USA) at room temperature and a constant voltage of 120 V. The gels were stained with silver stain plus. a dye kit (Bio-Rad). The molecular weight was determined using a marker Plus 2 pre-stained standards (Blue, Invitrogen). Native gel electrophoresis was carried out using a Bio-Rad 1000/500 power supply at 150 V for 1.5 h. The sample was loaded onto a 4–16% Bis-Tris gel Native PAGE (Invitrogen). The bands were visualized by staining with Coomassie Brilliant Blue G-250 (Bio-Rad).

#### 2.6.2. Effects of pH and Temperature on EAH Activity

The effect of temperature on tannase activity was determined over the temperature range of 20–80 °C (at pH 5). The thermostability of the enzyme was determined by 60 min of incubation over the same range monitoring the residual activity under assay conditions. Ellagitannin-acyl hydrolase activity as a function of pH was measured at 40 °C using punicalagin anomers as substrate. Citrate buffer in the pH range 3–6, phosphate buffer pH 7, and Tris-HCl buffer pH8 were used.

#### 2.6.3. Determination of Michaelis Constant (*Km*) and Maximum Reaction Velocity (*Vmax*)

The Michaelis constants (*Km*) and maximum reaction velocities (*Vmax*) were determined for the substrates punicalagin, methyl gallate (Sigma-Aldrich), and sugar beet arabinans (Megazyme). The concentration of substrates was 0.025 mM to 1.6 mM of punicalagin, 0.25 mM to 10 mM of methyl gallate, and 0.0625% to 2% of sugar beet arabinans. The constants were determined from the Lineweaver–Burk plots.

### 2.7. In-Gel Trypsin Digestion

The protein bands identified in fraction 3 of anionic exchange chromatography were manually excised and cut into 10 pieces of 1 mm and placed in 1.5 mL Eppendorf tubes. The gel pieces were detained using a solution of methanol: acetic acid: distilled water (10:1:9); then, they were dehydrated with acetonitrile and dried in a Speed Vac concentrator. The protein was reduced with a solution of 10 mM dithiothreitol (DTT) in 100 mM ammonium bicarbonate (NH_4_HCO_3_) for 1 h at 65 °C. DTT was removed, and the proteins were alkylated with 55 mM iodoacetamide in 100 mM NH_4_HCO_3_ for 1 h at room temperature in the dark. Proteins alkylated were then washed with 100 mM NH_4_HCO_3_, dehydrated with acetonitrile, and dried in Speed Vac. Tryptic digestion was carried out by adding porcine trypsin (Promega) in a ratio of 1 µg per 50 µg of protein and incubated at 37 °C overnight. The peptides were recovered by washing five times the gel fragments with 100 mM NH_4_HCO_3_. The solutions containing peptides were filtered through a 0.2 µm centrifugal membrane (Nanosep, PALL Corp, New York, NY, USA), dried in Speed Vac, and stored at −20 °C until LC-MS/MS analysis.

### 2.8. LC-MS/MS Analysis

Obtained peptides were dissolved in 20 µL of 0.1% formic acid and loaded into a Finnigan LTQ MS/MS system (Thermo Scientific) equipped with an electrospray ionization source. Six milliliters of every sample were injected in a HALO C18 column (Advanced Materials Technology, Wilmington, DE, USA), and peptides were separated using 0.1% of formic acid (solution A) and 80% of acetonitrile and 0.1% of formic acid (solution B) by a 90 min gradient of increasing solution B at a flow rate of 200 nL per minute. MS/MS ions spectra were recorded.

### 2.9. Protein Identification

The mass spectra were converted into mzXML files and then into peak list (PKL) files using the Trans-Proteomic Pipeline (Seattle Proteome Center, Seattle, WA, USA). The PKL files were then matched against the National Center for Biotechnology Information (NCBI) database containing Aspergillus niger proteins by using Mascot (Matrix Scientific, Boston, MA). Protein sequences were reversed to obtain the decoy database. Obtained proteins were statistically analyzed via ProteoIQ (PREMIER Biosoft, Palo Alto, CA, USA) by loading Mascot data and reversed sequences into the software (Version 1, DTU Health Tech, Athens, GA, USA).

### 2.10. Bioinformatic Analysis

Obtained aminoacidic sequences (identified as EAH1 and EAH 2) were submitted to the I-TASSER server for structural prediction. Models obtained were visualized using Discovery Studio Visualizer 2016 (BIOVIA). The amino acid sequences were also submitted to the NetNGlyc 1.0 server site (http://www.cbs.dtu.dk/services/NetNGlyc, accessed on 10 May 2018) [25] and NetOGlyc 4.0 server (http://www.cbs.dtu.dk/services/NetOGlyc/, accessed on 10 May 2018) [26] for the prediction of N-glycans and O-glycans linking sites. The interaction between enzyme and punicalagin was analyzed by submitting the amino acid sequence and the punicalagin structure (Human Metabolome Data Base number 05795) to PatchDock server (http://bioinfo3d.cs.tau.ac.il/PatchDock/, accessed on 10 May 2018) [27,28]. Hierarchic clustering analysis comparing the taninolitic enzymes produced by *A. niger* GH1 was conducted using MEGA version 6 by neighbor-joining analysis of Kimura-2 parameter distance estimates. The robustness of the tree was determined by bootstrap analysis (1000 replicates).

## 3. Results

### 3.1. Purification of EAH

The EAH was purified as described in the Materials and Methods section, and the results are shown in Table 1. The concentration of crude extract by lyophilizing resulted in the decrease in the enzymatic activity due to the concentration of salts and polyphenols that were present in the culture media. However, the High Prep Sephadex G-25 column allowed separating the protein from the polyphenols and salts, increasing the specific activity value (0.65 U/mg) and the purification fold (16.25-fold) in comparison to lyophilized extract. In the anionic exchange column, the EAH activity was identified in fraction number 3. The protein was unbounded to the column; probably the charges in the enzyme are negative and a cationic exchange chromatography is necessary. Notwithstanding, the purification fold enhanced 27.75-fold times and the specific activity reached 1.11 U/mg. In gel filtration chromatography, two fractions were identified with EAH activity: they were fraction 18 and fraction 23 (Table 1). In fraction 18, a low purification fold was reached (14.25-fold). On the other hand, in fraction 23, the highest values of specific activity (3.00 U/mg protein) and purification degree (75-fold times) were reached.

### 3.2. Molecular Weight of EAH

From the cell-free extract to gel filtration fractions, two bands of protein were identified on SDS-PAGE (Figure 1). Notwithstanding, after visualizing the fraction on SDS-PAGE, the same electrophoretic profile was identified, and the two bands observed from previous steps were visualized. The use of gel filtration chromatography did not allow the separation of the two bands visualized in the previous purification steps. In all purification steps, the two bands were visualized, and according to the molecular marker, the molecular weight was approximately 70 kDa for band 1 (identified as EAH 1) and 50 kDa for band 2 (EAH 2). After SDS-PAGE analysis, fraction 3 from anionic exchange chromatography was selected for the native PAGE analysis due to the high content of protein. In non-denaturing PAGE (Figure 2), the same two bands with the same molecular weight were visualized.

### 3.3. Effect and Temperature on EAH Activity

Since the two proteins could not be separated by gel filtration chromatography, fraction 3 obtained from anionic exchange chromatography was used to evaluate the effect of pH and temperature. Firstly, the pH was evaluated, and the results obtained demonstrated the highest relative activity value at pH 5 (Figure 3A). The enzyme showed 60% and 80% of relative activity at pH 3 and 4, respectively. On the contrary, at pH values of 6, 7, and 8, the enzyme exhibited relative activities under 20%. Figure 3B shows the results for the effect of temperature on EAH activity. The optimal temperature was 40 °C; however, over 60% of relative activity was kept at 50, 60, and 70 °C.

### 3.4. Kinetic Parameters

The kinetic properties of EAH were assayed, and the *Km*, *Vmax*, and the fit to the model for three substrates were determined (Table 2). Both identified proteins demonstrated the ability to catalyze the hydrolysis of punicalagin, methyl gallate, and sugar beet arabinans. The comparison of *Km* and *Vmax* values among the substrates showed the lowest values for punicalagin. The values were 0.053 mM (5.75 μg/mL) and 21.46 µM/min for *Km* and *Vmax*, respectively. This demonstrates the highest affinity of the enzyme for the punicalagin as substrate. The complexity of punicalagin probably caused the lower fit to the linear model (R^2^= 0.93). The enzyme has also affinity to methyl gallate as substrate showing a *Km* value of 6.66.mM (1.23 mg/mL) and a *Vmax* value of 128.20 µM/min. Sugar beet arabinans showed the lower values of *Km* (11.77 mg of arabinans/mL) but not for *Vmax* (662.25 µg/mL/min).

### 3.5. Molecular Properties of the Enzyme and In Silico Analysis

The proteins obtained in fraction 3 of anionic exchange chromatography were separated by SDS-PAGE, excised from the gel, digested by trypsin, and submitted for the HPLC-MS/MS analysis. The hierarchical clustering of the obtained sequences is shown in Figure 4. The two proteins (EAH 1 and EAH 2) obtained from *A. niger* GH1 were isoforms. The length for both proteins was 499 amino acids. The isoelectric point was 4.10 for EAH 1 and 4.24 for EAH 2. Some differences in the position of amino acids were identified. The data generated from the protein identification are shown in Table 3.

The comparison of both sequences with homologous proteins in the databases resulted in the identification of two α-l-arabinofuranosidase enzymes (Table 3 and Figure 4). The results showed identity up to 99% for both proteins and 100% sequence coverage. The sequence of the isolated enzyme was also compared to the enzymes produced by *A. niger* GH1 reported earlier [12,29,30,31]. It includes the tannase produced by submerged and solid-state fermentation and the pool of enzymes produced by the microorganism using pomegranate polyphenols. Figure 4 shows that the enzymes isolated in the present work were clustered into the clade of α-l-arabinofuranosidase from *A. niger*. The tannases produced under submerged and solid-state fermentation formed a separate clade, while the enzymes produced by *A. niger* GH1 using pomegranate polyphenols (β-glucosidase, Triacylglycerol lipase A, and phospholipase) form two clades separate from the tannases and the arabinofuranosidases.

The molecular weight obtained from SDS-PAGE informed that EAH 1 weighs 70 kDa and EAH 2 weighs 50 kDa. However, the data obtained from the sequence of both proteins showed similar molecular weight. The sequences were submitted to an analysis to identify the differences in molecular weight. The prediction of possible binding sites of N-glycans and O-glycans was carried out. Figure 5 shows the possible binding sites of N-glycans for EAH 1 and EAH 2. The red line is the threshold, and any potential crossing the default threshold of 0.5 represents a predicted glycosylated site. The O-glycan binding sites prediction is depicted in Table 4. All the results with score values up to 0.5 are shown.

### 3.6. Homology Modeling and Docking Analysis

The structure of both α-l-arabinofuranosidases was built using the I-TASSER server with the chain of 1WD3A as a template. The sequence identity between the template and target was 98.1% for EAH 1 and 93.8% for EAH 2. The coverage values were 96.6% for both proteins. As shown in Figure 6 and Figure 7, the proteins were organized into two domains (described above). According to I-TASSER analysis, the EAH 1 characteristics are: six helixes, five are in the first domain; it has 27 strands and 33 coils. On the other hand, the EAH 2 I-TASSER reported only 4 helixes, 28 strands, and 32 coils. After the analysis, no active site residues were obtained for both proteins.

The docking analysis of enzymes and punicalagin was carried out using the PATCHDOCK server. EAH 1 and EAH 2 were submitted as receptor molecules and the punicalagin (Human Metabolome Data Base number 05795) was submitted as a ligand. The punicalagin interacts with both domains present in the proteins (Figure 6 and Figure 7). However, differences in the ligand-binding sites were also identified. For the interaction between punicalagin and EAH 1, the amino acids PRO142, ASP,174, ASN202, TYR417, THR429, LYS430, GLN431, GLU434, ASP435, and TYR456 are involved. The ACE (Atomic Contact Energy) value for the interaction between EAH 1 and punicalagin was 10.27. The amino acids involved interact mainly with the phenolic rings from HHDP and with the two gallic acids attached to the gallagyl and linked to the hexose core at C-4 and C-6. EAH 2 interacts with punicalagin at the binding site composed of the following amino acids: PRO142, GLN303, ARG415, TYR417, ASN418, GLN431, ASP435, and TYR456. Despite more amino acids being involved in the interaction between protein and punicalagin, lower ACE (Atomic Contact Energy) values were obtained (−18.56), and the interaction was mainly with the gallagyl radical.

## 4. Discussion

### 4.1. The α-l-Arabinofuranosidase with Ellagitannin-Acyl Hydrolase (EAH) Activity

EAH was produced by SSF in Erlenmeyer flasks following the conditions described above. The fermentation was stopped after 18 h of culture. The time was defined according to data generated in previous works [9]. The enzymatic activity obtained in cell-free extract was 189 U/L. The activity obtained in the present work is lower than that reported by Buenrostro-Figueroa et al. [22], which reached 1400 and 1200 U/L using sugar cane bagasse and corn cobs, respectively, as solid supports and supplemented with pomegranate-husk polyphenols as an EAH inducer. They also used the *A. niger* GH1 strain, but the substrate provided to the enzyme in the EAH assay was total polyphenols from pomegranate husk. In the present work, the punicalagin anomers were used as the substrate for the enzymatic assay, and the release of ellagic acid was measured. It is clear that lower amounts of ellagic acid are present in the punicalagin structure; for that reason, lower EAH activity was obtained.

Information related to the expression of proteins with the ability to degrade tannins obtained from *A. niger* GH1 has been published previously. Mata-Gómez et al. [29] and Ramos et al. [30] reported a tannase with a molecular weight of 225 kDa. Renovato et al. [31] demonstrated different mass weights between two tannases from *A. niger* GH1 under submerged fermentation and solid-state fermentation. The proteins showed masses of 102 and 105 kDa for enzymes obtained in solid-state fermentation and submerged fermentation, respectively. The molecular weight for EAH was reported by Ascacio-Valdés et al. [20] and showed a mass range between 116 and 120 kDa. The enzyme is also produced by *A. niger* GH1.

The present study is the first report that purifies the enzyme capable of degrading ellagitannins. The study corroborated the presence of two forms of the same protein (named EAH 1 and EAH 2) found in both denatured and non-denatured PAGE. The molecular weight of EAH 1 (52.47 kDa) was similar to that of EAH 2 (52.63 kDa). Both enzymes showed the highest affinity for punicalagin as substrate, despite the bonds catalyzed, its positions, and the arrangement on the space making it more difficult to catalyze the reaction than sugar beet arabinans, and methyl gallate. For example, in punicalagin, there is an HHDP group linked to C-2 and C-3, and a gallagic acid linked to C-4 and C-6 (Figure 8). Meanwhile, the sugar beet arabinans are linked by bonds α1-2, 1-3, or 1-5. Moreover, using punicalagin as a substrate, the release of ellagic acid was quantified. This implies that the enzyme hydrolyzes two bonds for the release of ellagic acid, while for methyl gallate and sugar beet arabinans, the enzyme needs to break one bond for the release of one molecule of gallic acid and arabinose, respectively.

Punicalagin was used as an inducer for the production of EAH 1 and EAH 2. In addition, for the enzyme assay, punicalagin was used as substrate, and the release of ellagic acid was quantified. The molecule of punicalagin is composed of a sugar core (generally hexose), one molecule of hexahydroxidiphenic acid (HHDP) linked to the sugar core by ester bonds in position C-2 and C-3, and di-gallagyl linked to C-4 and C-6 by ester bonds [2]. Probably, the presence of bonds in C-2, C-3 and C-6 can induce the microorganism to produce the two isoforms of α-l-arabinofuranosidase due to the similarity of bonds that the enzyme can catalyze.

For that reason, in silico and docking analysis were carried out to obtain the binding sites where punicalagin interacts with both proteins (EAH 1 and EAH 2) and understand the interactions of the enzyme with a non-conventional substrate. Notwithstanding, an X-ray and/or SPR analysis is necessary to elucidate the interaction between the α-l-arabinofuranosidases and punicalagin.

### 4.2. Enzyme Characteristics

For the first time, we reported the degradation of ellagitannins by the action of an α-l-arabinofuranosidase. Two domains were identified in both proteins: the domain α-l-arabinofuranosidase B, catalytic (ArabFuran-catal) and the α-l-arabinofuranosidase B domain (AbfB). The ArabFuran-catal domain catalyzes the hydrolysis of α-1-2-, α-1-3-, and α-1-5-l-arabinofuranosidic bonds in arabinoxylans and L-arabinans [32]. Meanwhile, the AbfB domain hydrolyzes 1,5-a, 1,3-a, and 1,2-a linkages in both oligosaccharides and polysaccharides, which contain terminal non-reducing L-arabinofuranoses in the side chains [33].

The α-l-arabinofuranosidase (AFase) is a glycosyl hydrolase enzyme and has been classified into many families (3, 10, 43, 51, 54, and 62 of the CAZy glycosyl hydrolases) according to the amino acid sequence similarities [34]. The enzyme catalyzes the breakdown of terminal and side-chain bonds of arabinofuranosides in arabinans (α-1, 5- linked arabinofuranosides) and the arabinosyl within arabinogalactans and arabinoxylans (C-2 and C-3 positions) [35,36]. In that sense, AFases have displayed many different substrate specificities within any given glycosyl hydrolase family.

The arabinofuranosidases are capable of degrading a wide range of substrates including natural polymeric polysaccharides and chromogenic PNP-glycosides. For synthetic substrates, the α- and β- forms of arabinofuranoside, arabinopyranoside, xylopyranoside, galactopyranoside, and glucopyranoside, among others, have been employed, all of them linked to the PNP. On the other hand, sugar beet arabinan, debranched arabinan, arabinogalactan, and arabinoxylan, among others, are the natural substrates degraded by arabinofuranosidases [37].

The α-l-arabinofuranosidases have been reported as a wide substrate catalyzer. For example, its activity against monoterpenyl α-l-arabinofuranosylglucosides from grape has been demonstrated, liberating monoterpenyl β-D-glucosides and arabinose [38]. The α-l-arabinofuranosidases also release ferulic acid and p-cumaric acid bounded to C-5 in the arabinofuranosyl side group present in arabinoxylans [39]. Even arabinofuranosidases from *A. niger* are capable of releasing hexoses such as galactose in a similar way to arabinose [40].

The differences in the molecular weight of both α-l-arabinofuranosidases are due to the presence of glycans in the structure of the enzymes. For that reason, the in silico analysis of N- and O-glycans binding sites was carried out. N-glycosylation is known to occur on asparagine, which occurs in the Asn-Xaa-Ser/Thr stretch (where Xaa is any amino acid except Proline). While this consensus tripeptide (N-glycosylation sequon) may be a requirement, it is not always sufficient for asparagine to be glycosylated. The sequence of EAH 2 reveals a change in position 202 where the amino acid threonine was identified, while in EAH 1, in the same position, asparagine was identified.

The results show that EAH 1 has two possible sites where N-glycans can be attached to the protein: the sites are the amino acids number 83 and 202. In the EAH 2 sequence, only one possible site was obtained, and it is in the amino acid number 83 coinciding with one of the sites obtained for EAH 1. Similar to N-glycans results, more sites for O-glycans binding were found in protein EAH 1. Ten positive positions were found in EAH 1, while in EAH 2, only four positive positions were identified. However, there are some coincidences in both proteins such as the amino acids in positions 62, 272, and 355. The in silico identification of N- and O-glycans helped us to predict the glycosylation sites in both proteins and justify the differences in mass weight identified by SDS-PAGE described previously. However, an analysis for the identification of both N- and O-glycans is necessary to elucidate their structures.

## 5. Conclusions

The purification, identification, and characterization of an enzyme produced by *A. niger* GH1 capable of degrading ellagitannins were conducted in the present work. The identification of the enzyme by HPLC-MS/MS reveals the presence of two α-l-arabinofuranosidases with similar lengths, molecular weight, and isoelectric point. The evaluation of kinetic constants reveals a high affinity of the enzymes for punicalagin as substrate. The differences in mass weight identified in SDS-PAGE analysis were justified by the glycosylation degree (N- and O-glycans). The I-TASSER server allowed us to obtain the homology 3D models of both proteins. In addition, the docking analysis reveals the interaction of different amino acids with punicalagin. Both proteins were able to degrade ellagitannins and to release ellagic acid.

## Figures and Tables

**Figure 1 foods-12-00903-f001:**
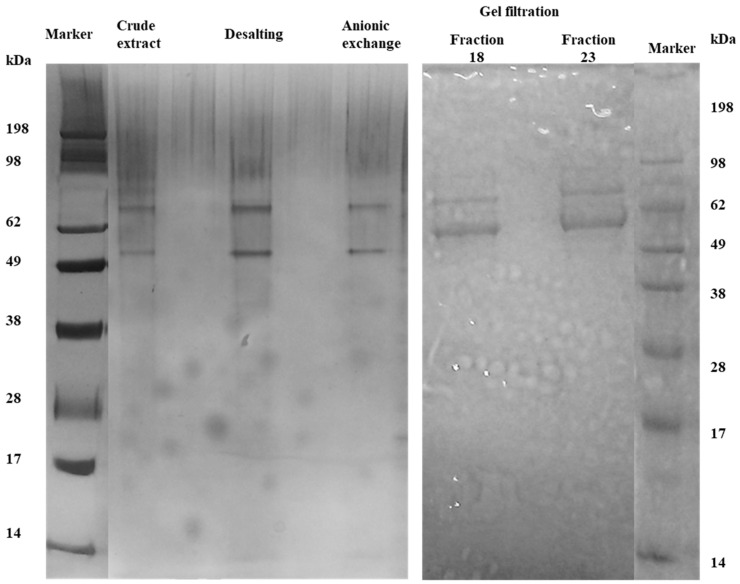
SDS-PAGE of EAH showing the two isoforms of EAH obtained from *Aspergillus niger* GH1.

**Figure 2 foods-12-00903-f002:**
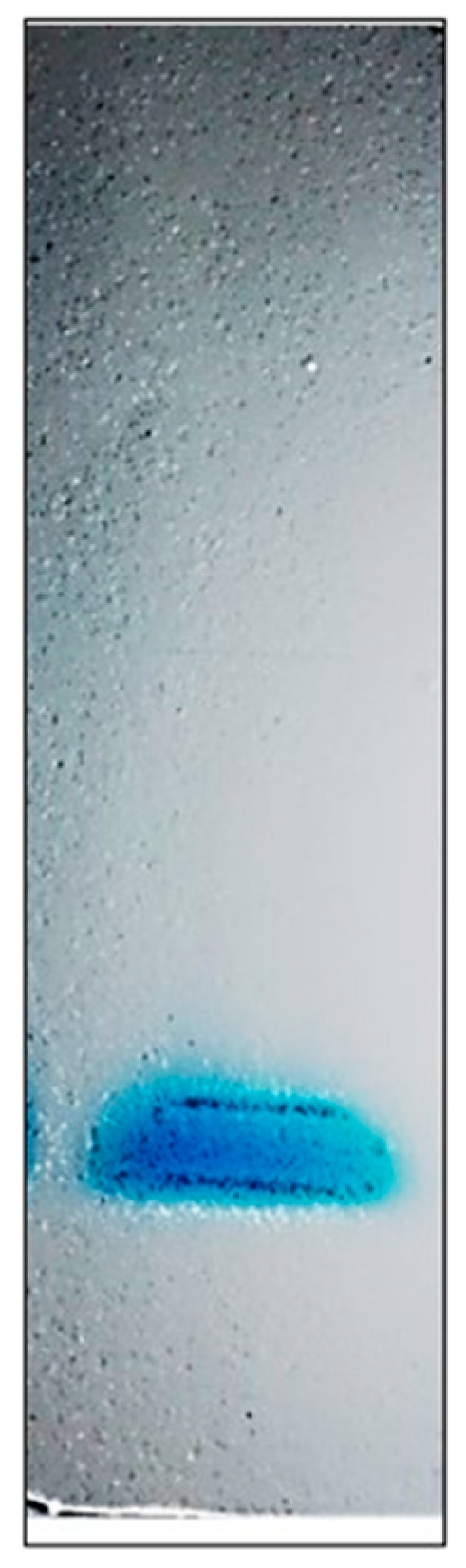
Non-denaturing PAGE of EAH in fraction 3 from anionic exchange chromatography.

**Figure 3 foods-12-00903-f003:**
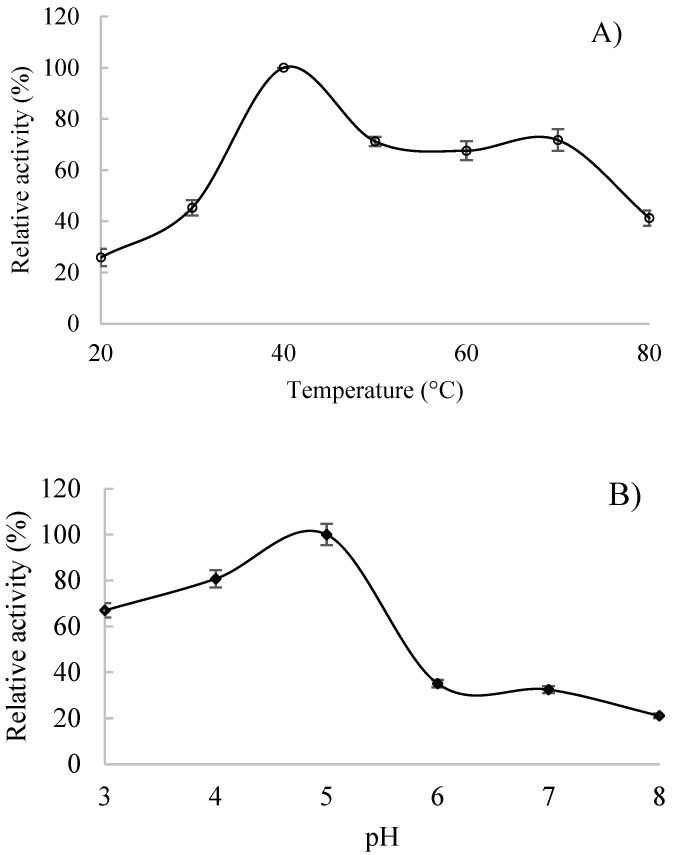
Physical characterization of α-l-arabinofuranosidase. (**A**) Effect of temperature on EAH activity and (**B**) Effect of pH on EAH activity. Mean and standard deviations were plotted.

**Figure 4 foods-12-00903-f004:**
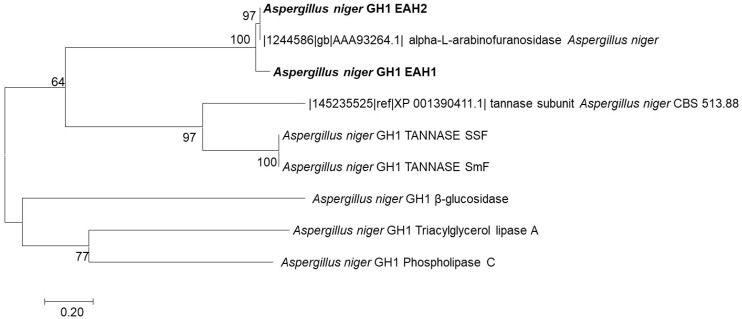
Hierarchical clustering of the taninolitic enzymes produced by *A. niger* GH1.

**Figure 5 foods-12-00903-f005:**
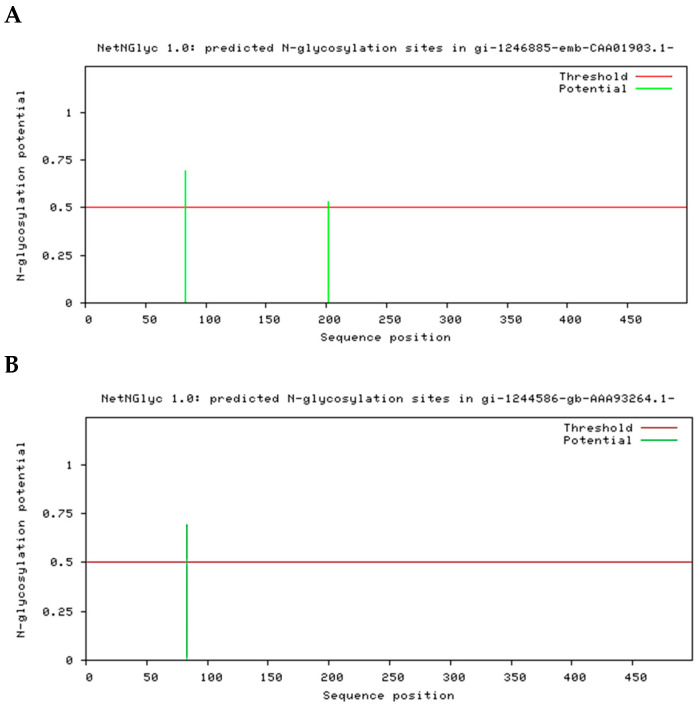
Predicted binding sites of N-glycans on the identified proteins secreted by *A. niger* GH1. (**A**) represents the graphic for EAH 1 and (**B**) are the results obtained for EAH 2.

**Figure 6 foods-12-00903-f006:**
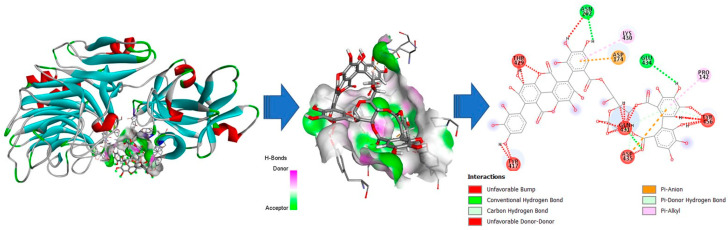
Docking analysis for the interaction between EAH 1 and punicalagin. The ligand-binding site residues are: PRO142, ASP,174, ASN202, TYR417, THR429, LYS430, GLN431, GLU434, ASP435, and TYR456.

**Figure 7 foods-12-00903-f007:**
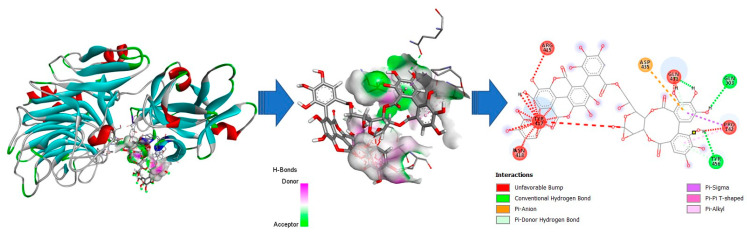
Docking analysis for the interaction between EAH 2 and punicalagin. The ligand-binding site residues are: PRO142, GLN303, ARG415, TYR417, ASN418, GLN431, ASP435, and TYR456.

**Figure 8 foods-12-00903-f008:**
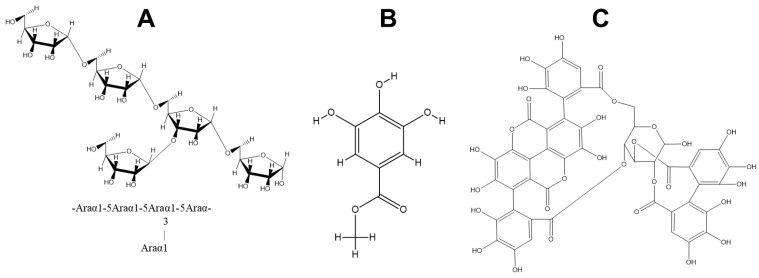
Structures of arabinan (**A**), methyl gallate (**B**), and punicalagin (**C**) used as substrates for the enzymatic assays.

**Table 1 foods-12-00903-t001:** Purification strategy for extracellular EAH produced by *Aspergillus niger* GH1.

Purification Step	Volume (mL)	Protein (mg)	Total Activity (U)	Specific Activity (U/mg)	Purification Fold	Yield (%)
Lyophilized extract	20	68.84	2.96	0.04	1.00	100.00
Sephadex G-25 column	5	0.63	0.41	0.65	16.25	13.85
Anionic Exchange	2	0.09	0.10	1.11	27.75	3.37
Gel filtration Fraction 18	2	0.07	0.04	0.57	14.25	1.35
Gel filtration Fraction 23	2	0.01	0.03	3.00	75.00	1.01

**Table 2 foods-12-00903-t002:** Kinetic parameters in the hydrolysis of punicalagin, methyl gallate, and sugar beet arabinans by EAH.

Substrate	*K_m_* (mM)	*V_max_* (µM/min)	R^2^
Punicalagin	0.053	21.46	0.93
Methyl gallate	6.66	128.20	0.95
Sugar beet Arabinans	11.77 *	662.25 **	0.97

***** mg/mL; ** μg/mL/min.

**Table 3 foods-12-00903-t003:** Characteristics of EAH 1 and EAH 2 proteins.

Parameter	EAH 1	EAH 2
Protein length (amino acids)	499	499
Protein weight (kDa)	52.47	52.63
Isoelectric point	4.10	4.24
Domains	ArabFuran-catal (20–334)	ArabFuran-catal (20–333)
	AbfB (352–493)	AbfB (352–493)
Homologous ID	gi|1246885|emb|CAA01903.1|	gi|1244586|gb|AAA93264.1|
Query cover (%)	100	100
Identity (%)	99	100
Sequence name	α-l-arabinofuranosidase B [*Aspergillus niger*]	α-l-arabinofuranosidase [*Aspergillus niger*]

**Table 4 foods-12-00903-t004:** O-glycans binding sites prediction.

EAH 1			EAH 2		
Amino Acid Position	Score	Comment	Amino Acid Position	Score	Comment
38	0.568	Positive	62	0.744	Positive
44	0.513	Positive	152	0.539	Positive
62	0.776	Positive	272	0.617	Positive
67	0.531	Positive	355	0.594	Positive
187	0.516	Positive			
272	0.616	Positive			
351	0.632	Positive			
355	0.828	Positive			
360	0.791	Positive			
361	0.549	Positive			

## Data Availability

Data supporting the reported results can be requested directly by email contact to the corresponding author. For the amino acid sequences; http://www.cbs.dtu.dk/services/NetNGlyc, accessed on 10 May 2018. The interaction between enzyme and punicalagin was analyzed by submitting the amino acid sequence and the punicalagin structure (Human Metabolome Data Base number 05795) to PatchDock server (http://bioinfo3d.cs.tau.ac.il/PatchDock/, accessed on 10 May 2018).

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
