# Peer review of "Production of a Fungal Punicalagin-Degrading Enzyme by Solid-State Fermentation: Studies of Purification and Characterization"

_foods, 2023, doi:10.3390/foods12040903_

Round 1

Reviewer 1 Report

The present manuscript by Aguilar-Zárate et al described the production , purification and characterization of a fungal punicalagin-degrading enzyme by solid state fermentation

My concerns

The paper need an extensive English editing by a native English speaker as it has been written mostly in misleading or wrong way

The paper contains countless typos, misspelling, spacing errors suggesting the authors did not read the manuscript carefully before submission

The all the figures showed in the paper are really of poor quality e.g., figure 1, figure 3, figure 4, figure 5 and the docking too and also the authors did not label the residues on the docking

Section 3.3 molecular properties of EAH is not convincing at all; for example < In non-denaturing  PAGE (Figure 2), the same two bands with the same molecular weight were visualized> how did you visualize these band and compared their molecular weight? And I can not see two bands.

Why the SDS-PAGE in figure 2 showed only two bands even in the crude extracts?and was the gel for both crude extracted and purified protein after gel filtration chromatography run similantously? I believe no because the gel seems compiled of two separate gels, this is a major reliability concern.

Line 204-205> The EAH was produced by SSF in Erlenmeyer flasks following the conditions previously 204 described, please add a reference for “previously described”

In kinetic parameter section 3.5, the authors exaggerate comparing the Km and Vmax of the three substrates used though it cannot be performed considering sugar beet arabanin was dissolved as %w/v solution while others as molarity, so the results are not comparable

Figure 3: Physical characterization of α-L-arabinofuranosidase. A) Effect of temperature on EAH activity. 271 B) Effect of pH on EAH activity. Mean and standard deviations were plotted. Why different names are being used for this protein frequently and standard errors are not visible.

The authors should display the structure of all the three substrates

The results of protein identification using trypsin digestions and LC/MS-MS are not adequately described

Author Response

The paper need an extensive English editing by a native English speaker as it has been written mostly in misleading or wrong way

R: The language was revised by academic editors from the Autonomous University of San Luis Potosí.

The paper contains countless typos, misspelling, spacing errors suggesting the authors did not read the manuscript carefully before submission.

R: The manuscript was re-read, and the errors were corrected.

The all the figures showed in the paper are really of poor quality e.g., figure 1, figure 3, figure 4, figure 5 and the docking too and also the authors did not label the residues on the docking

R: The figures were improved as the reviewer suggested. The docking figures were restructured to be clearer.

Section 3.3 molecular properties of EAH is not convincing at all; for example < In non-denaturing  PAGE (Figure 2), the same two bands with the same molecular weight were visualized> how did you visualize these band and compared their molecular weight? And I can not see two bands.

R: Denaturing SDS-PAGE were visualized by staining the gels with silver stain. The non-denaturing PAGE was stained with Coomasie brilliant blue G-250. The figure 1 was improved and the two bands now can be seen.

Why the SDS-PAGE in figure 2 showed only two bands even in the crude extracts? and was the gel for both crude extracted and purified protein after gel filtration chromatography run similantously? I believe no because the gel seems compiled of two separate gels, this is a major reliability concern.

R: The SDS-PAGE in the figure 2 shows 2 bands because A. niger GH1 produced two enzymes that allowed to degrade the punicalagin provides in the culture medium as carbon source. The samples crude extract, desalting and anionic exchange were run simultaneously. The samples for the gel filtration chromatography were run in a separate gel.

Line 204-205> The EAH was produced by SSF in Erlenmeyer flasks following the conditions previously 204 described, please add a reference for “previously described”

R: It was changed ‘previously’ by ‘above’ since the idea refers to the methods depicted in the manuscript.

In kinetic parameter section 3.5, the authors exaggerate comparing the Km and Vmax of the three substrates used though it cannot be performed considering sugar beet arabanin was dissolved as %w/v solution while others as molarity, so the results are not comparable.

R: To be comparable, the Km value for arabinans was recalculated and expressed as mg/mL. Also, in the text was included the corresponding values for punicalagin and methyl gallate.

Figure 3: Physical characterization of α-L-arabinofuranosidase. A) Effect of temperature on EAH activity. 271 B) Effect of pH on EAH activity. Mean and standard deviations were plotted. Why different names are being used for this protein frequently and standard errors are not visible.

R: The name of the protein was homogenized, and the plots were edited to make visible the standard deviation.

The authors should display the structure of all the three substrates

R: The structures of the substrates were included in the manuscript.

The results of protein identification using trypsin digestions and LC/MS-MS are not adequately described

R: The depiction of the results was improved.

Reviewer 2 Report

The paper is well written, but before publication, I suggest making some improvements as folow:

1. Add in the introduction more data regarding the used microorganism_ Aspergillus niger GH1.

2. In the Materials and methods section, please state providers for all reagents/media/filters, etc. 

3.  The discussion section is missing! This part is the most immportant. Indeed some subsection of the Results section have discussions, but not very developed. Eather make a "Results and discussions" section  or a "Discussion" section with in deep evaluation of the obtained results in comparisoon with the scientific literature on the topic. 

4. Please reformulate more relevant and specific conclusions.

5. Please double-check the paper for typos and text format. 

Author Response

The paper is well written, but before publication, I suggest making some improvements as folow:

  1. Add in the introduction more data regarding the used microorganism_ Aspergillus niger GH1.

The information was added.

  1. In the Materials and methods section, please state providers for all reagents/media/filters, etc. 

The information was included.

  1. The discussion section is missing! This part is the most immportant. Indeed some subsection of the Results section have discussions, but not very developed. Eather make a "Results and discussions" section  or a "Discussion" section with in deep evaluation of the obtained results in comparisoon with the scientific literature on the topic. 

The discussion section was included.

  1. Please reformulate more relevant and specific conclusions.

The conclusions were modified.

  1. Please double-check the paper for typos and text format. 

The typos and text format were revised and corrected.

Round 2

Reviewer 1 Report

Though authors addressed several issues raised in my first report.

I am still not convinced regarding the gel pictures and marker labeling on both sides and also paper still need careful check for typos and spacing errors.

Author Response

The document was double checked for typos and spacing errors. We regret the reviewer is not convinced of the gel pictures but these are the images we have available.

Reviewer 2 Report

The paper is more clear now and has a better flow. 

Author Response

Thank you for your revision and comments.
